

# The Fluxgate Magnetometer of the Low Orbit Pearl Satellites (LOPS): overview of in-flight performance and initial results

Ye Zhu[1, 2, 3], Aimin Du[1, 2], Hao Luo[1, 2], Donghai Qiao[4], Ying Zhang[1], Yasong Ge[1, 2], Jiefeng Yang[3], Shuquan Sun[1], Lin Zhao[1], Jiaming Ou[1], Zhifang Guo[1], Lin Tian[1, 2]

[1]Key Laboratory of Earth and Planetary Physics, Institute of Geology and Geophysics, Chinese Academy of Sciences, Beijing, 100029, China
[2]College of Earth Science, University of Chinese Academy of Sciences, Beijing, China
[3]Shanghai Engineering Centre for Microsatellites, Chinese Academy of Science, Shanghai, China
[4]School of Electronic and Information Engineering, Soochow University

*Correspondence to*: H. Luo (luohao06@gmail.com)

**Abstract.** The Low Orbit Pearl Satellite series consists of six constellations, with each constellation consisting of three identical micro-satellites which line up just like a string of pearls. The first constellation of three satellites were launched on September 29, 2017, with an inclination of ~ 35.5° and ~ 600 km altitude. Each satellite is equipped with three identical Fluxgate Magnetometers (FGM), which measure the in-situ magnetic field and its low frequency fluctuations in the Earth's

low altitude orbit. The triple sensor configuration enables separation of stray field effects generated by the spacecraft from the ambient magnetic field [e.g. Zhang et al., 2006]. This paper gives a general description of the magnetometer about the instrument design, calibration before launch, in flight calibration, as well as the in-flight performance and initial results. Unprecedented spatial coverage resolution of the magnetic field measurements allow for investigating the dynamic processes and electric currents of ionosphere and magnetosphere, especially for the ring current and equatorial electrojet (EEJ) during

both geomagnetic quiet conditions and storms. It could be important for studying the method to separate their contributions of the M-I current system.

## 1 Introduction

Magnetic fields are fundamental elements in characterizing the Earth's environment. Accurate and high spatial coverage of the magnetic field vector measurements along the orbits (35.5° inclination, 600 km altitude) of LOPS allow for separation of

temporal and spatial of the magnetic field and hence beneficial to study the magnetospheric and ionospheric magnetic features of the external field at mid-to-low latitude, which is important for establishing a high precision geomagnetic model [e.g. Hulot et al., 2015; Olsen et al., 2016]. In particular, with the simultaneous multiple magnetic field observations at mid-to-low latitude, the ring current, especially the partial ring current, as well as the equatorial electrojet at different local time could be studied in a great detail. In addition, the dynamic change of the south Atlantic anomaly under different geomagnetic

activities could also well monitored with the help of dense magnetic field observation coverage of local time at mid-to-low latitude.

The magnetic field intensity at low Earth orbit is in the range from ~20000 nT to ~60000 nT. In addition, some scientific research of the physical processes such as the geomagnetic pulsations require the magnetic field resolution as high as 0.1 nT [e.g. Sutcliffe et al., 2000]. These conditions raise high requirements for the low Earth orbit magnetic measurements such as

the satellite platforms, instrumentation design, and data calibrations. There were no global high-precision measurements of the Earth's magnetic field until the launch of the OGO-2 satellite in 1965, though this satellite only measured the magnetic intensity at altitudes from 400 km to 1510 km [Cain and Langel, 1971]. The MAGSAT is the first global magnetic vector survey satellite, which operated for about six month form November 1979 to April 1980 [Mobley et al., 1980]. It was about 20 years after the MAGSAT mission the more recent and high precision global magnetic satellite observations became

available: the ørsted satellite [Olsen, 2007], CHAMP [Maus, 2007], and SAC-C [Stauning, 2003] carried nearly the same instrumentation and provided over a decade unique geomagnetic data sets, which were used for establishing a lot of geomagnetic models [e.g. Olsen et al., 2002, 2004, 2010; Sabaka et al., 2004; 2012; Maus et al., 2007; Finlay et al., 2016]. The three-satellites mission SWARM from the European Space Agency (ESA), launched on November 22, 2013, provide not only the global magnetic field measurements but also the east-west gradient of the magnetic field with the help of two

flying side by side spacecraft with a separation in longitude of about 1.4o [Friis-Christensen et al., 2006]. The SWARM mission provides the best ever survey of the geomagnetic field and its spatial and temporal evolution. During the past decades, the academic-commercial consortium provided magnetic field data from the Iridium constellation of more than 70 communication satellites to the geospace science community. Although without the magnetic cleanliness, the iridium engineering magnetometer data also provided important information for studying and sensing the global field-aligned current

[Anderson et al., 2000; Anderson et al., 2002, Waters et al., 2001; Anderson et al., 2008]. By using multipoint the magnetic measurements from the three Space Technology 5 (ST5) satellites [Slavin et al., 2008], Le et al. [2009], for the first time, separated the temporal and spatial variations in field-aligned current perturbations in low-Earth orbit on time scales of ~10 sec to 10 min. There are three identical fluxgate magnetometers equipped on LOPS, with sensors 1 and 3 mounted at the tips of two 1.5m booms on each side of the spacecraft, as shown in **Figure 1**. The sensor 2 is mounted at the middle of boom at

the sensor 3 side. Without magnetic cleaning of the satellite platform, the magnetic environment conditions could not be satisfied for the demand of the geomagnetic measurements before magnetic cleanliness program were made. Previous studies have shown that the spacecraft stray field caused by magnetic material or generated by the platform currents could be detected and removed to be below the threshold of the scientific requirement by using a difference or gradient method based on dual sensors measurements [e.g. Zhang et al., 2006, 2007; Auster et al., 2008; Ludlam et al., 2008; Pope et al., 2011].

With three sensors on LOPS, the separation of the ambient and the stray magnetic field (both DC and AC field) becomes possible. The magnetic investigation of the spacecraft body, the payloads, and the solar panels were carefully examined before the launch of each satellite. In addition, in flight stray field determination is carried out based on several different methods. We will give detailed description of the stray field corrections for both pre and in flight periods in section 4.2.



The Star Imager (SIM), which determines the attitude of the satellite with high accuracy, is mounted in the satellite body.

This configuration, of course, will lead to the errors of the three Euler angles (which describe the rotation between the coordinate system of the magnetometer and the SIM) to some extent due to that the boom which connects the magnetometer and the satellite body is not total rigid and it will vibrate in orbit [Olsen et al., 2003].

The rest of this paper is organized as follows: in Section 2 we give the magnetometer instrument description which include the fluxgate sensor and the sensor electronics. The magnetometer calibration for both pre and inflight is the main topic of the

Section 3. In Section 4, we give the initial scientific results. The summary is given in Section 5.

## 2 Instrument Description

### 2.1 Overview

The fluxgate magnetometers are the most common magnetometers which are used for space magnetic field measurements. The LOPS flux magnetometer consists of a vector compensated three axis fluxgate sensor unit and a digital electronics on a

single printed circuit board. The electronic box comprises three sensor electronics boards, the Data Processing Unit (DPU) board and Power Control Unit. Both the sensors and electronic onboard the LOPS benefit from the heritage of the magnetometers of the Venus Express (Zhang et al., 2006) and onboard the THEMIS (Auster et al., 2008). The main instrument parameters are listed in Table 1.

### 2.2 Fluxgate Sensors

The fluxgate sensor consists of a sense coil surrounding an inner excitation coil that is closely wound around a highly permeable, low noise, and good offset stability soft magnetic ring-cores. The material used for the magnetic ring-cores are similar to those used on Venus Express (Zhang et al., 2006) and Equator-S and has been tested for a strict procedure such as the vibration and temperature cycling.

The sense coil consists of three mutually perpendicular component coils and has the triaxial concentric shape in order to

make sure the three measured components are the magnetic information from a spatial location with different directions.

The feedback coil is also composed of three mutually perpendicular coils. The feedback circuit generate additional three-component magnetic fields in real time in order to ensure the uniformity and stability of the generated magnetic field, each direction coil is composed of two sets of parallel coils.

A continuous repeating cycle electromagnetic signal drove by the excitation coil is monitored by the sense coil with the

principal frequency twice that of the excitation signal frequency, and whose strength and phase orientation vary directly with the external-field magnitude and polarity.

The real sensor picture and the structure design diagram are shown in **Figure 2a** and **2b**.

### 2.3 Sensor Electronics

The sensor electronics consists of excitation module circuitry, sense signal acquisition module circuitry, feedback module
circuitry, and the temperature module circuitry. The excitation module, which is composed of the excitation signal portion
generated in the FPGA and MOS drive amplification circuit, is used to generate the excitation signal required for the
excitation coil. The schematic of the excitation module circuit is shown in **Figure 3**.

In the excitation module circuitry, the FPGA generates two 9.6 kHz square wave signals with opposite phases, which are
amplified by the power amplifier circuit and transmitted to the excitation coil. In addition, the excitation circuit also includes
a circuit that forms LC resonance with the internal excitation coil of the probe.

The sensing signal acquisition module circuitry is used to collect the sensing signal. The output signal from the sense coil of
the sensor is firstly amplified by the instrumentation amplifier, and then sampled and converted by the ADC and transmitted
to the FPGA. The block diagram is shown in **Figure 4**.

The feedback module circuitry is designed to generate a feedback signal to the feedback coil to form a feedback magnetic
field to compensate the external magnetic field. It consists of DAC circuit and voltage-controlled constant current source
circuit. The 12-bit high-resolution and high precision DAC contains the anti-interference circuits. The classic Howland
current source circuit built by the operational amplifier is selected to be the voltage-controlled constant current source. When
the resistance is matched, the output resistance tends to be infinite. At this time, the voltage signal is converted into a linear
current signal, which is independent of the load and the operating frequency, that is, the output current is a constant, and
resulting a constant compensation magnetic field. Its block diagram is shown in **Figure 5**.

The temperature module is designed to collect the voltage divider value of the sensor and the thermistor in the electronics as
the temperature value. It is composed of a series voltage divider circuit, ADC chip, and an FPGA. The temperature
measuring circuit consists of a thermistor and a voltage dividing resistor. The ADC chip is responsible for collecting the
voltage dividing value of the thermistor and the FPGA is responsible for controlling the ADC chip for acquisition, and
encapsulating the collected temperature data into the scientific data packet and then sent to the payload controller.

The electronics with the functionalities descripted above placed on a shared board is shown in **Figure 6**. The board area is
about 120 cm2 and the total power consumption is 0.5 W. The total mass (including the harness) is 1.6 kg.

### 3 Instrument Calibration before launch

### 3.1 Ground Calibration

All the fluxgate magnetometers were well calibrated before launch. The parameters of fluxgate magnetometers to be
determined during ground calibration include the scale factor, linearity, frequency response, orthogonality of the triaxial
sensor, time stability of the sensor offset, noise of sensor and electronics, temperature stability of sensor offset, and etc. In
this section, we will give the detailed calibration processes for determining calibration parameters mentioned above.



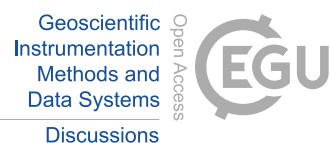

In the absence of an external magnetic field, the strength of the magnetic field given by the magnetometer is considered as the fluxgate magnetometer offset (b), which is independent systematically on sensor and electronics temperature. There is a proportional relationship between the output of each axis of the magnetometer and the real magnetic field. The proportional coefficient is called the scale factor, which is usually different for each axis and independent of the external conditions. The scale factor can be expressed by the following diagonal matrix:

$$\boldsymbol{K}_{SF} = \begin{bmatrix} s_X & & \\ & s_Y & \\ & & s_Z \end{bmatrix} \tag{1}$$

The elements on the diagonal indicate the scale factor of each axis. There is a certain deviation in the direction between the axes of the magnetometer and the ideal Cartesian coordinate system, which leads to the non-orthogonal of in the three sensitive axes of the magnetometer, as shown in **Figure 7**. Where X, Y, Z is the axes of the magnetometer. Establishing an orthogonal coordinate system with Z axis of the magnetometer as the z axis, with Y axis in the y-O-z plane, and $B_x$, $B_y$, and $B_z$ are projections of the magnetic field strength on axes of orthogonal coordinate system, respectively. The

direction error between the magnetometer sensitive axis and the orthogonal coordinate system axis can be represented by three error angles. The angle between the projections of the X-axis in the x-O-y plane and the x-axis is α, the angle between the X-axis and the x-O-y plane is β, and the angle between the Y-axis and the y-axis is γ. The projection of the magnetic field strength on the three axes of the magnetometer is:

$$\begin{bmatrix} B_X \\ B_Y \\ B_Z \end{bmatrix} = \boldsymbol{K}_{NO} \begin{bmatrix} B_x \\ B_y \\ B_z \end{bmatrix} \tag{2}$$

Where

$$\boldsymbol{K}_{NO} = \begin{bmatrix} \cos\alpha\cos\beta & \sin\alpha\cos\beta & \sin\beta \\ & \cos\gamma & \sin\gamma \\ & & 1 \end{bmatrix} \tag{3}$$

The relationship between the magnetometer output $\boldsymbol{B}_m$ and the real magnetic field strength $\boldsymbol{B}$ in the ideal sensor orthogonal coordinate system is:

$$\boldsymbol{B}_m = \boldsymbol{K}_{sf}\boldsymbol{K}_{NO}\boldsymbol{B} + \boldsymbol{b} \tag{4}$$

Then $\boldsymbol{B}$ can be expressed as:

$$\boldsymbol{B} = \boldsymbol{K}\left(\boldsymbol{B}_m - \boldsymbol{b}\right) \tag{5}$$

Where:





$$\boldsymbol{K} = \left(\boldsymbol{K}_{sf}\boldsymbol{K}_{NO}\right)^{-1} = \begin{bmatrix} s_X\cos\alpha\cos\beta & s_X\sin\alpha\cos\beta & s_X\sin\beta \\ & s_Y\cos\gamma & s_Y\sin\gamma \\ & & s_Z \end{bmatrix} \tag{6}$$

If we set:

$$\boldsymbol{K}^{\mathrm{T}}\boldsymbol{K} = \begin{bmatrix} d & e & f \\ e & g & h \\ f & h & i \end{bmatrix} \tag{7}$$

Calculating the modulus of both sides of Equation (5), we can obtain:

$$
\begin{aligned}
& dB_{mx}B_{mx} + 2eB_{mx}B_{my} + 2fB_{mx}B_{mz} + gB_{my}B_{my} + 2hB_{my}B_{mz} + iB_{mz}B_{mz} + \\
& \left(-2db_x - 2eb_y - 2fb_z\right)B_{mx} + \left(-2eb_x - 2gb_y - 2hb_z\right)B_{my} + \left(-2fb_x - 2hb_y - 2ib_z\right)B_{mz} + \\
& db_xb_x + 2eb_xb_y + 2fb_xb_z + gb_yb_y + 2hb_yb_z + ib_zb_z - \\
& \left|\boldsymbol{B}\right|^2 = 0
\end{aligned}
\tag{8}
$$

Equation (8) can be sorted as the following linear equation:

$$\begin{bmatrix} B_{mx}B_{mx} \\ 2B_{mx}B_{my} \\ 2B_{mx}B_{mz} \\ B_{my}B_{my} \\ 2B_{my}B_{mz} \\ B_{mz}B_{mz} \\ -2B_{mx} \\ -2B_{my} \\ -2B_{mz} \\ 1 \end{bmatrix}^{\mathrm{T}} \begin{bmatrix} d \\ e \\ f \\ g \\ h \\ i \\ \boldsymbol{K}^{\mathrm{T}}\boldsymbol{K}\boldsymbol{b} \\ \boldsymbol{b}^{\mathrm{T}}\boldsymbol{K}^{\mathrm{T}}\boldsymbol{K}\boldsymbol{b} - \left|\boldsymbol{B}\right|^2 \end{bmatrix} = 0 \tag{9}$$

If the external magnetic field keeps constant, and one can rotate the magnetometer to obtain enough attitudes and Equation (9) becomes linear equations and $\boldsymbol{K}^{\mathrm{T}}\boldsymbol{K}$ and offset $\boldsymbol{b}$ can be calculated. Performing the Cholesky decomposition and taking the triangular matrix to get $\boldsymbol{K}$. From Equation (6):

$$\begin{cases} K_{11}^{2} + K_{12}^{2} + K_{13}^{2} = s_X^{2} \\ K_{22}^{2} + K_{23}^{2} = s_Y^{2} \\ K_{33} = s_Z \end{cases} \tag{10}$$

Where $s_X > 0$ and $s_Y > 0$, $\boldsymbol{K}_{SF}$ can be calculated. From Equation (6) again we can get:

$$\begin{cases} s_Y\sin\gamma = K_{23} \\ s_X\sin\beta = K_{13} \\ s_X\sin\alpha\cos\beta = K_{12} \end{cases} \tag{11}$$





We can calculate the $\alpha, \beta, \gamma$ under the condition that $\alpha, \beta, \gamma$ are small angles close to zero.

In addition to the above calibration processes, to get a sufficient statistics, the offset should be measured by sensor rotation in a weak field as often as possible, typically in the beginning and end of each calibration campaign. Table 2 summarized the calibrated linear parameters of the scaling factor, offset, and the orthogonality angle:

The test of the dependency of the magnetometer stability on temperature was performed in a temperature control box in which the temperature varied from -60° to 60°. The sensor electronics was mounted inside the temperature control box and the sensor was placed in a Helmholtz coil, in which the Earth's magnetic field was compensated to be a weak field with a factor of $10^4$. After the test the stability of the fluxgate magnetometer is less than 30 pT/°C for all the sensor axes.

The instrument noise was tested in both a Shielding bucket and the natural environment. **Figure 8** shows the results of the
noise test in a Shielding bucket. Panel (a) shows of time series of 80 seconds of magnetic intensity measured by the fluxgate magnetometer. The corresponding FFT spectra is shown in panel (b). As can be seen in panel (b), the noise is about 3 pT/√Hz at 1 Hz. The test results in a natural environment was shown in **Figure 9**. The noise is about 10 pT/√Hz at 1 Hz, which is slightly higher than that in a Shielding bucket. The dependency of the sensor and electronics noise on temperature from 0°C-60°C were also tested. The noise varies from 20 pT/√Hz at 0°C to 10 pT/√Hz at 20°C and to 30 pT/√Hz at 60°C.

**3.2 Magnetic survey of the spacecraft**

Since the LOPS was not designed to make a strict magnetic cleanliness of the spacecraft, a careful investigation of the spacecraft for both DC and AC field is needed before launch. Measurements indicate average values of ~900 nT and ~2000 nT of the AC field at two outboard sensors and the inboard sensor, respectively. There are about 10 nT of dynamic interferences (DC field) at two outboard sensors and ~50 nT at the inboard sensor. These values, though not very accurate,
are quite important as the reference for the inflight calibration of the magnetometers.

**4 Inflight Calibration: techniques and preliminary results**

After the launch of the LOP satellites, we performed the inflight calibrations, which consists of two categories, with one the spacecraft dynamic interferences (AC field) which was generated by the electronic current of the spacecraft and the other the static interference (DC field) which was generated by the hard iron and soft iron material onboard the spacecraft. In this
section, we will simply introduce the processes of the calibration techniques of the two interferences and give the preliminary results.

**4.1 Inflight calibration of the spacecraft dynamic interferences**

**Figure 10** shows an overview of the raw data of magnetic field observations in the VFM coordinate along with the attitude and orbit information from LOPS-1 from 1200 to 1600 UT, January 1, 2018. The top six panels shows the *Bx*, *By*, and *Bz*
component from sensors 1 and 3, respectively. The bottom five panels shows the three Euler angle (pitch, yaw, and roll),



which indicates the satellite attitude, and the orbit (longitude and latitude). As we can see in **Figure 9**, there are three time intervals in which strong dynamic interferences in magnetic field vector data for both sensor 1 and sensor 3 are observed. The three time intervals coincide with the time periods in which the three Euler angles equal to zero. It is noted that the dynamic interferences in sensor 1 are stronger than those in sensor 3. This may be due to that the sensor 1 are nearer to the

satellite antenna, though the two sensors are the same 1.5 meters away from the satellite platform body. Compared to the strong dynamic interferences the three time period, it is rather small and in the in the in-flight calibration processes we exclude the strong interferences period.

There are several transient signals that are sourced by the spacecraft, such as the antenna effects, the rotation effects of the platform, the Solar panel effect, and the electric system of the spacecraft. It should be noted that the spatial gradient of the

magnetic field sourced by the spacecraft at the three sensors is obviously larger than the natural magnetic field signal at low-altitude orbit of the Earth. Therefore, the differences of the magnetic field among the three sensors are caused only by the spacecraft dynamic interferences at a fixed time as long as the three sensors were well calibrated before launch and the offsets keep stable at orbit.

$$\boldsymbol{B}_{D12}(t) = \boldsymbol{B}_{S1}(t) - \boldsymbol{B}_{S2}(t) \tag{12}$$

$$\boldsymbol{B}_{D13}(t) = \boldsymbol{B}_{S1}(t) - \boldsymbol{B}_{S3}(t) \tag{13}$$

$$\boldsymbol{B}_{D23}(t) = \boldsymbol{B}_{S2}(t) - \boldsymbol{B}_{S3}(t) \tag{14}$$

Where $\boldsymbol{B}_{S1}$, $\boldsymbol{B}_{S2}$, and $\boldsymbol{B}_{S3}$ are the magnetic field sourced by the spacecraft at each sensor. $\boldsymbol{B}_{D12}$, $\boldsymbol{B}_{D13}$, and $\boldsymbol{B}_{D23}$ are the differences of the magnetic field measured by the three sensors and they are only a function of the spacecraft system effect. They contain information about all of the changes in the spacecraft field. Although it is difficult to determine the exact

magnetic field sourced by the spacecraft at each sensor, we can identify this signal according to the $\boldsymbol{B}_{D12}$, $\boldsymbol{B}_{D13}$, and $\boldsymbol{B}_{D23}$ in the MAG data. In the processes of the dynamic interference correction, the $\boldsymbol{B}_{D12}$, $\boldsymbol{B}_{D13}$, and $\boldsymbol{B}_{D23}$ are the basis of the method to identify the dynamic transient events sourced by the spacecraft. Once the dynamic events are identified, the effect that those dynamic events have on the measured field should be determined and corrected to the data to minimize the effect. After careful examination, we found the magnitude of the differences of the two outboard sensors $|\boldsymbol{B}_{d13}|$ substantially smaller than

that of the differences between the sensor 2 and sensor $|\boldsymbol{B}_{d23}|$, which is usually an order smaller, indicating that spacecraft dynamic field at the outboard sensor is considerably smaller than that of the inboard sensor. This attenuation is so remarkable that a significant amount of the dynamic interferences sourced by the spacecraft, especially during the intervals in which the spacecraft attitude changes gradually (for example during the interval 1245-1345 UT in **Figure 10**), are negligible at the two outboard sensors. However, in spite of the reduction, some transient dynamic events should also be

calibrated at the two outboard sensors.

**Figure 11** shows the original magnetic field measurements in the VFM coordinate system, the corresponding detrended (in blue) and the calibrated time series ( in red) onboard LOPS-1 in a half orbit during which the strong interferences are absent. Due to the strong background magnetic field, the dynamic interferences cannot be well examined in the raw data. Therefore, the detrended time series, which are shown below the corresponding original data, were obtained by subtracting the 120





seconds smoothed time series. A sawtooth signal can be seen in all three magnetic field components. It is possible that the sawtooth signal is associated with the loading current of the satellite. We will give a detailed description of this signal and its origin in a subsequent paper. In this paper, we briefly introduce how to diminish this signal and get a reasonable background magnetic field. We first make a smoothing spline fitting to the data series by selecting reasonable fitting parameters and then subtract the fitting curve by the original data to get the high-frequency component. First order difference is applied to the

high-frequency component. Then we will set the threshold to calibrate the data according to the first order differences. After this calibration, we add together the calibrated high-frequency component and the low-frequency one (the fitting data series) to obtain the final calibrated time series.

### 4.2 Inflight calibration of the spacecraft static interferences

Once the dynamic interferences fields were corrected in the spacecraft coordinates, we can estimate the static spacecraft field.

Unlike the static spacecraft field of Venus Express, which remains constant throughout the orbit [Pope et al., 2011] and could be estimated using a modified Davis-Smith method [Leinweber et al., 2008], the static spacecraft field of a low-altitude satellite is approximately composed of two parts, with one is a constant generated from the hard iron material and the other is induced magnetic field of the soft iron material and is proportional with the background magnetic field. The magnetic field from recent CHAOS-7 geomagnetic model [Finlay et al., 2020] were used to estimate the static spacecraft

field. Since the CHAOS-7 model were derived with the data only from dark regions, where the IMF Bz field averaged over the previous two hours was positive and the IMF By was less than + 3nT (N. hemisphere) or greater than -3 nT (S. Hemisphere) are used. The LOPS magnetic field data used to estimate the static field were also selected in the same criteria with the CHAOS-7 model.

Though the magnetometers onboard LOPS were calibrated on the ground, the parameters (scale factor, linearity,

orthogonality of the triaxial sensor, and the offset) of the magnetometers may change to some extent. Therefore, in the estimation of the static field, we also consider those parameters to be determined. The calibration model can be expressed as follows:

$$B_{mea} = E_{ma} E_{no} \left( E_{sf} \left( B_{earth} + K_i B_{earth} + B_p \right) + B_{ns} \right) \tag{15}$$

Where each variable are listed in the following table

| Parameters of magnetometer | | | | Static spacecraft interference | |
|---|---|---|---|---|---|
| non-orthogonality | Scale factor | Error of misalignment | offset | Hard-iron mag | Soft-iron Coff |
| $E_{no}$ | $E_{sf}$ | $E_{ma}$ | $B_{ns}$ | $B_p$ | $K_i$ |

The equation (15) can be rewrote as:

$$B_{mea} = K B_{earth} + b \tag{16}$$



Where:

$$K = E_{ma}E_{no}E_{sf} + E_{ma}E_{no}E_{sf}K_i \tag{17}$$

$$b = E_{ma}E_{no}E_{sf}B_p + E_{ma}E_{no}B_{ns} \tag{18}$$

$B_{mea}$ and $B_{earth}$ are the measurements of the magnetometer and the model values of CHAOS-7. Solve the linear equation (16), we can obtain the **K** and **b**:

$$
\begin{bmatrix} B_{mea,x} \\ B_{mea,y} \\ B_{mea,z} \end{bmatrix} = \begin{bmatrix} B_{mod,x} & B_{mod,y} & B_{mod,z} & 0 & 0 & 0 & 0 & 0 & 0 & 1 & 0 & 0 \\ 0 & 0 & 0 & B_{mod,x} & B_{mod,y} & B_{mod,z} & 0 & 0 & 0 & 0 & 1 & 0 \\ 0 & 0 & 0 & 0 & 0 & 0 & B_{mod,x} & B_{mod,y} & B_{mod,z} & 0 & 0 & 1 \end{bmatrix} \begin{bmatrix} K_{1,1} \\ K_{1,2} \\ K_{1,3} \\ K_{2,1} \\ K_{2,2} \\ K_{2,3} \\ K_{3,1} \\ K_{3,2} \\ K_{3,3} \\ b_x \\ b_y \\ b_z \end{bmatrix} \tag{19}
$$

**Figure 12** summarize the estimating processes.

**4.3 Preliminary results and comparison with geomagnetic models**

After the correction of the dynamic and static spacecraft interferences, we can obtain the Earth's natural magnetic vector. Since the observations between LOPS and SWARM are neither time synchronous nor at the same altitude, we just make a comparison between LOPS observations-based geomagnetic model and CHAO-7 model. The LOPS data-based geomagnetic model was stablished by using the Gauss Spherical Analysis based on the vector measurements of 12 satellites from the time period 1st April to 30th April, 2018. Data selection criteria for establishing the model is the same as CHAOS-7 model. In this simple model, we do not consider the geomagnetic secular variation since we only use data in a month. A more comprehensive model with magnetic field data from SWARM, LOPS, and CSES will be established in the near future in a sequent paper. **Figure 13** shows the three magnetic field components and the total intensity (up to spherical harmonic degree n=20) at 600 km altitude calculated from the geomagnetic model established based on the LOPS data. As we can see in this figure, the main features of the geomagnetic main field at the altitude 600 km are clearly shown. **Figure 14** shows the mean values distribution of the magnetic field residuals (after removing the core, crustal, and large-scale magnetospheric magnetic contributions calculated from CHAOS-7 model) in the Hammer-Aitoff map projection from time period 1st April to 30th April, 2018. Data with geomagnetic quiet period ($Kp \leq 3^0$) were selected to make this distribution. As we can see in Figure 12, despite being quiet periods, the magnetic residuals which describes the external field can be clearly seen in all three components as well as the total intensity. The most remarkable characteristics in the distribution of X component residual





can be seen in **Figure 14a**. The X component residual at most of the area covered by LOPS shows negative values except at the Indian Ocean. There is a clear negative narrow band near the equator, which may indicates the Equatorial Electrojet [e.g. Yamazaki and Maute, 2017]. Similar feature can also be seen in F component residual since the Equatorial Electrojet effect is the most remarkable signature at low-latitude at 600 km. The Y component residual (**Figure 14b**) shows different features,

with positive and negative values appearing alternately. This distribution is possible due to the inter-hemisphere field-aligned currents at middle latitudes [e.g. Fukushima, 1994; Lühr et al., 2015]. The Z component residual also shows the positive and negative values alternately and it has different features at northern and southern hemisphere. The explanation for this distribution remains unclear.

In order to examine the magnetic power spectra of the vector field calculated from the LOPS data-based geomagnetic model,

in **Figure 16**, we present the Lows-Mauersberger spherical harmonic power spectra for both LOPS-based model and CHAOS-7 model, in which the Gauss coefficients of LOPS model were obtained during the period 1$^{st}$ April to 30$^{th}$ April, 2018. It is shown that the spectra for the two models decrease steadily from n=1 to 13. From degree n=1 to n=9, the spectra of the two models agree well. After that, the two spectra show differences, with the power spectra of LOPS model slightly larger than that of CHAOS-7. It is possible that the power spectra of LOPS with degree n > 9 include the magnetic field

ingredient of Large-scale F-region currents since the spatial scale and the mean amplitude of the magnetic field generated by the currents is overlapped by the core field with degree n=6-13 [see Figure 3 of Olsen et al., 2012]. In addition, we believe that the power spectra also contain the magnetic field ingredient of EEJ though the spatial scale of the magnetic field generated by EEJ is smaller than the core field with degree n < 14. As we know, the EEJ are located at altitude of about 100-150 km [e.g. Yamazaki and Maute, 2017], which is below the LOPS orbit (600 km). Therefore, in the simple spherical

harmonic analysis, the magnetic field generated by the EEJ is treated as the internal field and may not be isolated with the core and crustal field. The strict isolation of the EEJ field will be taken into account by modelling the EEJ and/or data selection in the comprehensive model setting up in the future.

We also examined the ability to capture the EEJ by several specific orbits. Due to the limited amount of calibrated data, only small number of valid events are identified until now. We followed the operation of [Alken and Maus, 2007; Lühr et al.,

2004], and estimated the core field, crustal field, and field of the magnetosphere by CHAOS-7 geomagnetic model [Finlay et al., 2020; Olsen et al., 2006]. After subtracting the magnetic field of other sources, the measurement was considered to include only the magnetic effects from the ionospheric current system, i.e. the Sq and EEJ effects. The results of the residual north magnetic component for different orbits are shown with different coloured lines in **Figure 17**. Though a lot of small fluctuations appear in each orbit measurement, the clear EEJ signature with about 20 nT dip at the zero dip latitude could be

seen in the thick black line (the averaged value for all the orbits). The asymmetry of the residual magnetic field may be attributed to the spacecraft trajectory, which covers several local times for a specific orbit. Detailed analysis of the EEJ currents captured by our magnetometer measurements will be presented in a sequent paper [Tian and Luo, 2021, in submission].



In order to make a comparison with CHAOS-7 model, we show the residual distribution of the LOPS magnetic field vector and intensity data during the period of 1$^{st}$ April to 30$^{th}$ April, 2018 (shown in **Figure 15**). The residual were obtained by removing the core, crustal, and the magnetospheric field as given by CHAOS-7 model. The average values of the residuals are -0.47 nT, -0.12 nT, -1.48 nT, and -1.12 nT for X, Y, Z components and magnetic intensity, respectively. The absolute deviations are 14.65 nT, 20.50 nT, 22.29 nT, and 18.61 nT for the three components and intensity, respectively. It should be noted that the orbit altitude of the LOPS (~ 600 km) is slightly higher than that of SWARM B (~ 530 km). Therefore the

CHAOS-7 model values at ~600 km are in fact the magnetic field upward continuation. In addition, it also should be noted that the residuals may also contain the magnetic field ingredient generated by the Sq, EEJ, and large-scale F-region currents since those currents are not modelled in the CHAOS-7 model and data in the dayside were not excluded in the residuals calculations.

## 5 Summary

Benefitted from the good inherited of the development of the ring cores, the sensor design and the technology of the electronics of the fluxgate magnetometers, the magnetometers onboard the LOPS provide the capability to monitor the Earth's magnetic field at low-altitude orbit after remove both the dynamic and static interferences sourced by the spacecraft. Dozens of measurements (15 satellites, 45 magnetometers) at low-altitude orbit lead to the challenge of the data calibration and analysis method to achieve some important scientific results. On the other hand, with unprecedented spatial coverage of

the low-altitude and inclination of the magnetic field measurements as well as the data accumulates, it also presents opportunities to study the magnetic field of the Earth in a great detail, especially the electric current system at mid-low latitude such as the ring current and the EEJ, which is quite important for separation of the internal and external magnetic field for establishing more accurate geomagnetic models.

## Acknowledgements

This work was supported by National Key R&D Program of China (Grant no. 2018YFC1503806), the Strategic Priority Research Program of Chinese Academy of Sciences (Grant No. XDB41010304), Beijing Municipal Science and Technology Commission (Grant No. Z191100004319001) and the National Natural Science Foundation of China (41874080, 41874197).



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





| Table 1. The instrument parameters of the magnetometers onboard LOPS. | |
|---|---|
| Range | ±65000nT |
| Resolution | 10 pT |
| Noise | ≤ 10pT/√Hz @ 1Hz |
| Data Rate | 1Hz, 16Hz, 32Hz, 128Hz |
| Power consumption | 5.8 W (Max) |
| Mass | |
|    Sensor | 100g |
|    Electronics (including harness) | 1600g |
| Dimensions | |
|    Sensor | 67.7±1mm × 68±1mm × 42±1mm |
|    Electronics | 165mm × 123mm × 53mm |


| Table 2. The parameters of the linear calibration for sensors 1-9 | | | | | | | | | |
|---|---|---|---|---|---|---|---|---|---|
| | S1 | S2 | S3 | S4 | S5 | S6 | S7 | S8 | S9 |
| X scaling factor | 0.84333 | 0.8539 | 0.8352 | 0.8518 | 0.8473 | 0.8677 | 0.8395 | 0.8391 | 0.8426 |
| Y scaling factor | 0.79294 | 0.7908 | 0.7853 | 0.8022 | 0.8005 | 0.7948 | 0.7959 | 0.8036 | 0.7971 |
| Z scaling factor | 0.91217 | 0.8909 | 0.9142 | 0.9212 | 0.9148 | 0.9166 | 0.9103 | 0.9050 | 0.9081 |
| Offset X (nT) | -101.14 | 828.55 | -1176.1 | 680.79 | 607.02 | 787.48 | 777.44 | -1790.8 | -588.04 |
| Offset Y (nT) | -533.40 | -1752.4 | 510.16 | -790.96 | 1492.6 | 263.75 | 576.95 | 870.79 | -682.28 |
| Offset Z (nT) | -535.53 | 597.49 | 614.37 | 1555.1 | 335.93 | -42.76 | -902.61 | 518.03 | -475.83 |
| Orth (XY) (°) | -0.5782 | -0.2593 | -0.1568 | -0.1981 | -0.3848 | 1.0255 | -0.2795 | 0.3674 | -0.5730 |
| Orth (YZ) (°) | 0.8835 | -0.4482 | -0.6651 | 0.1421 | 0.3241 | 0.9584 | 0.8743 | 1.1245 | -0.5186 |
| Orth (XZ) (°) | -0.3219 | 00.9139 | -0.1635 | 0.3689 | -0.1577 | 0.9584 | 0.3863 | -0.1538 | 0.3199 |



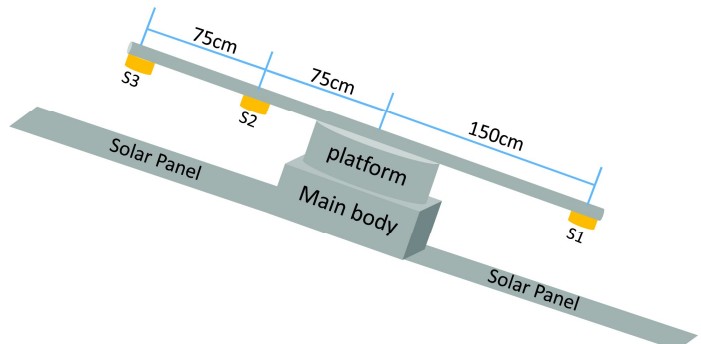

**Figure 1: Schematic diagram of the installation positions of the three magnetometer sensors on board the satellite.**



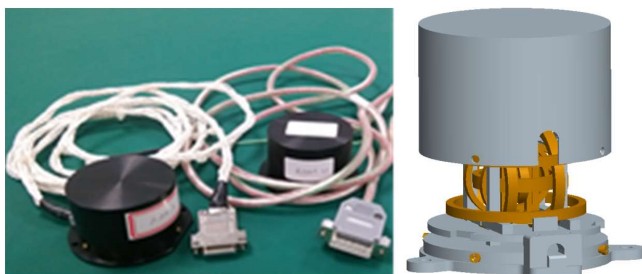

**Figure 2: (a) The real sensor picture and (b) the structure design diagram of the sensor.**



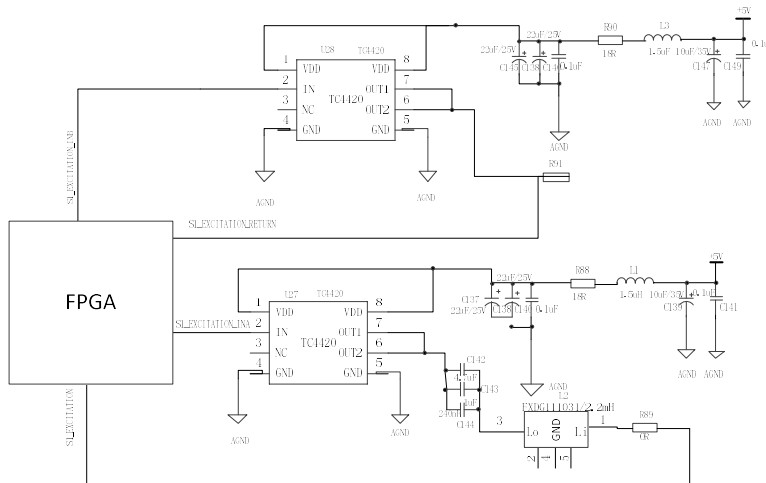

**Figure 3: The schematic of the excitation module circuit.**


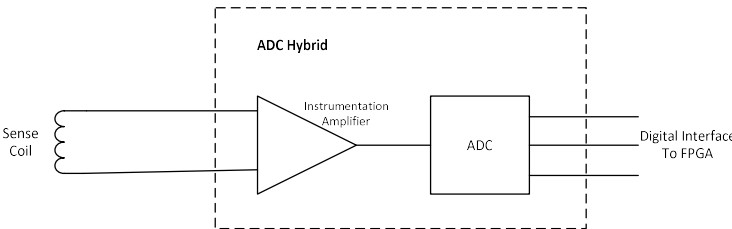

**Figure 4: The block diagram of the sensing signal acquisition module.**

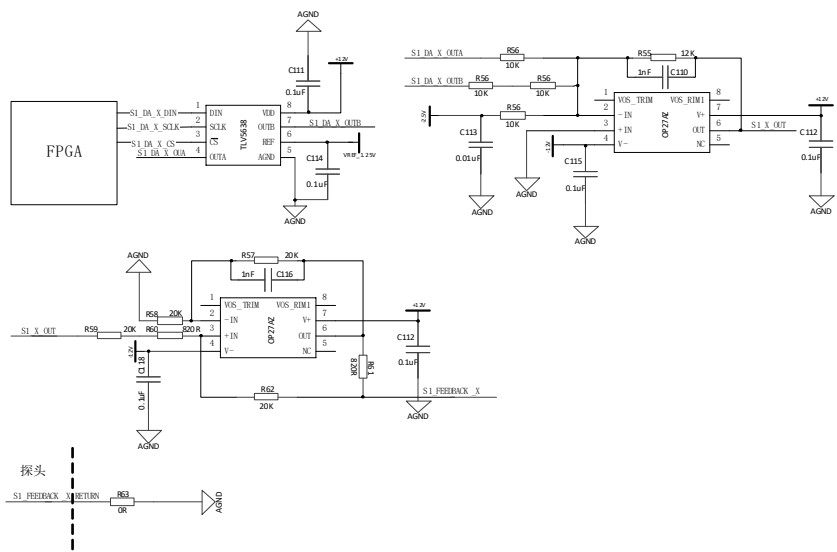

Figure 5: The block diagram of the feedback module circuitry.

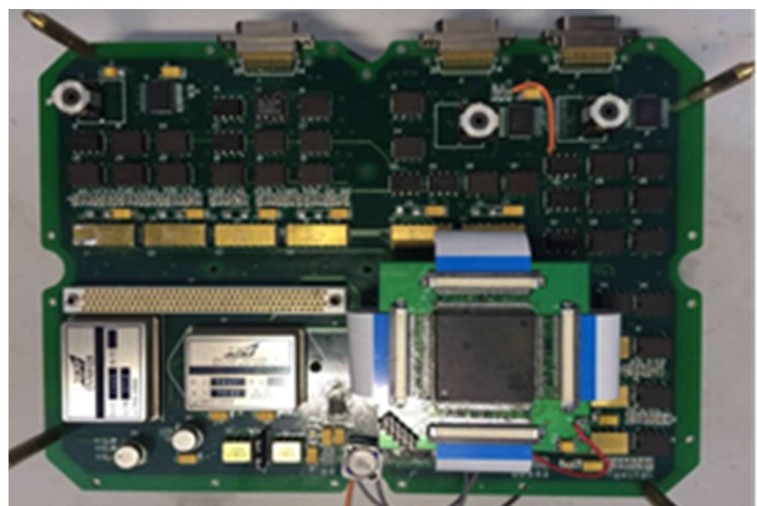

Figure 6: The fluxgate magnetometer electronics placed on a share board.





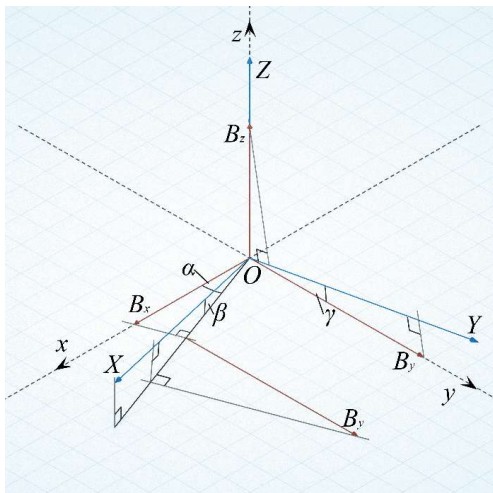


**Figure 7: Schematic diagram of non-orthogonal error of three-axis fluxgate magnetometer.**

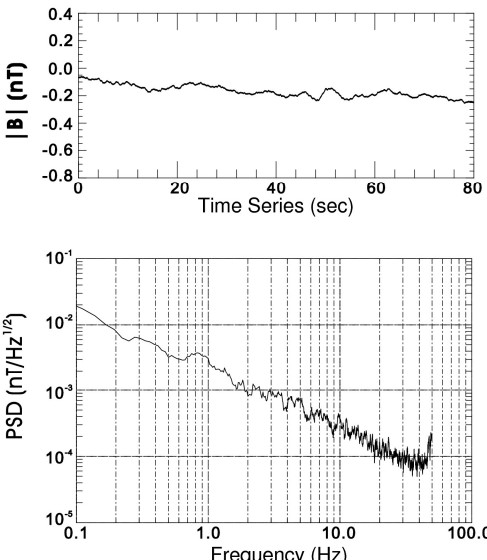

**Figure 8: The noise test in a Shielding bucket: (a) The time series of the magnetic field intensity. (b) The corresponding FFT spectra of the time series.**





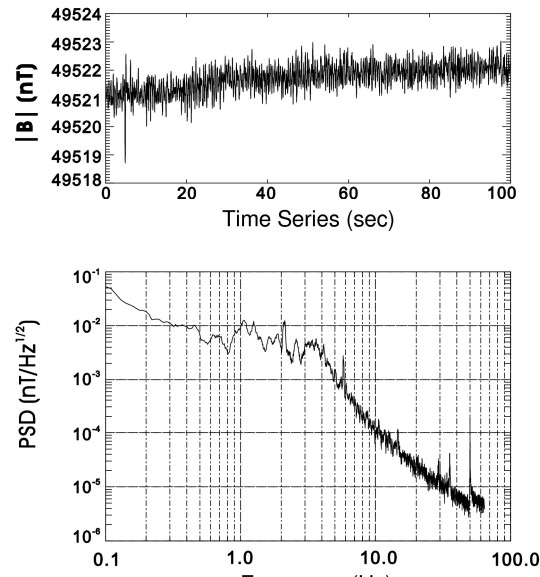


**Figure 9: The noise test in a natural environment: (a) The time series of the magnetic field intensity. (b) The corresponding FFT spectra of the time series.**

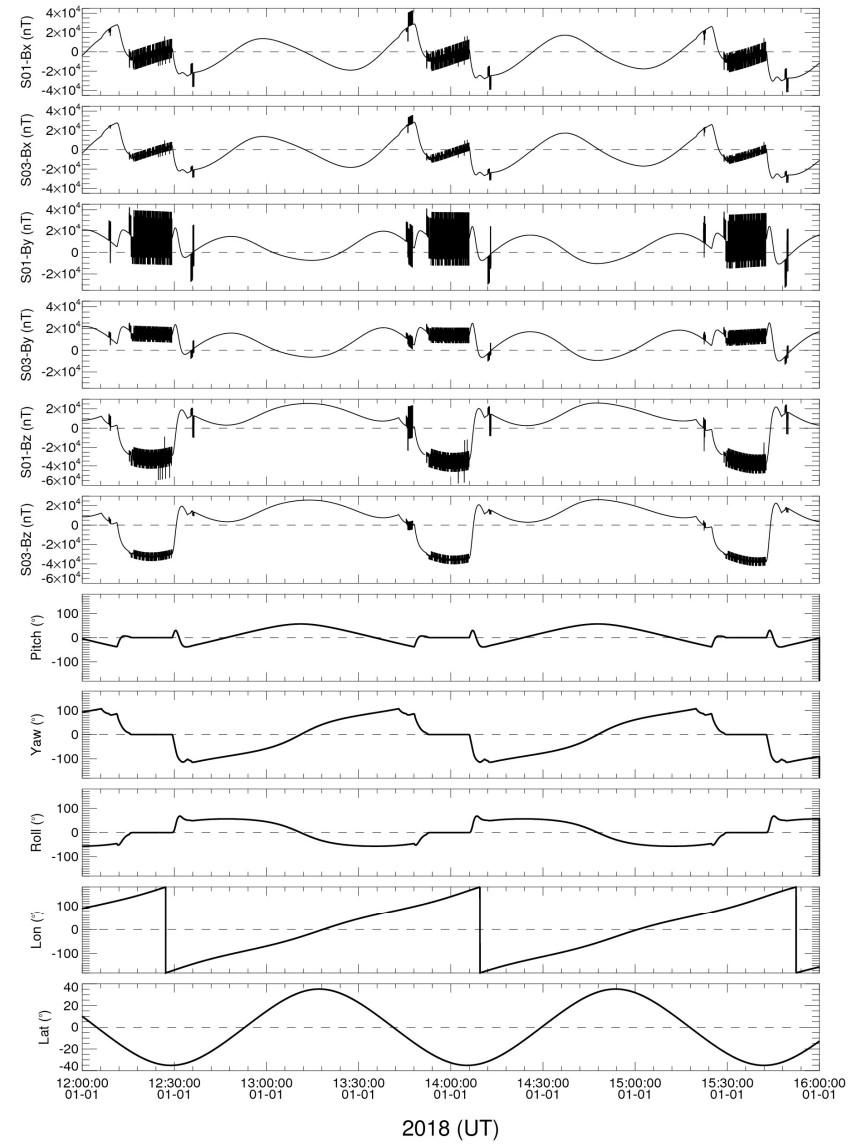

**Figure 10: Overview of the raw data of LOPS-1, including the magnetic data from sensor 1 and sensor 3, the satellite attitude and orbit.**


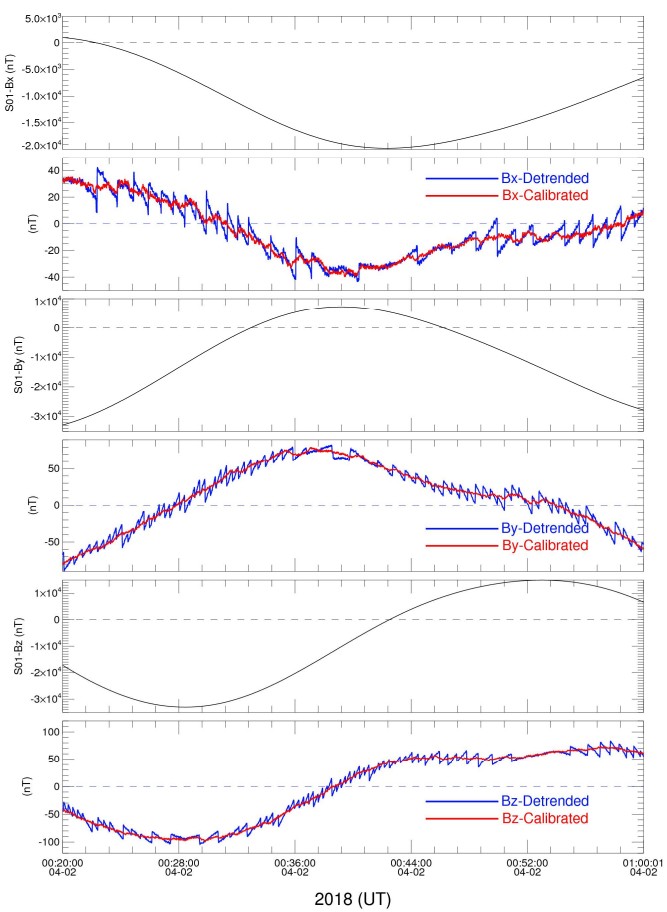

**Figure 11: The overview of the LOPS-1 data before (blue) and after dynamic interference correction (in red).**





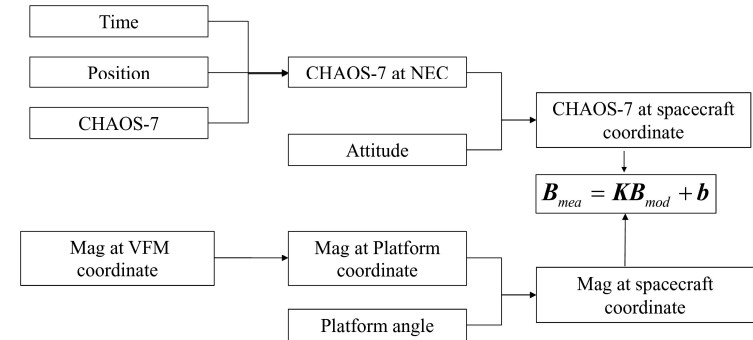

**Figure 12: The processes of the static spacecraft interferences estimation based on CHAOS-7 model.**

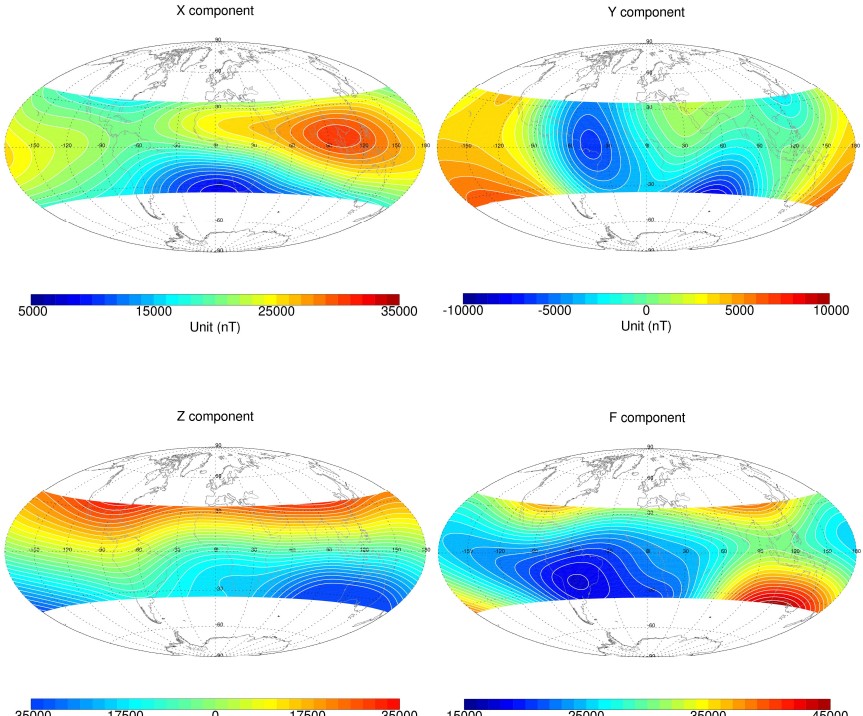

**Figure 13: The three magnetic field components and the total intensity at 600 km altitude calculated from the geomagnetic model**
**established based on the LOPS data.**



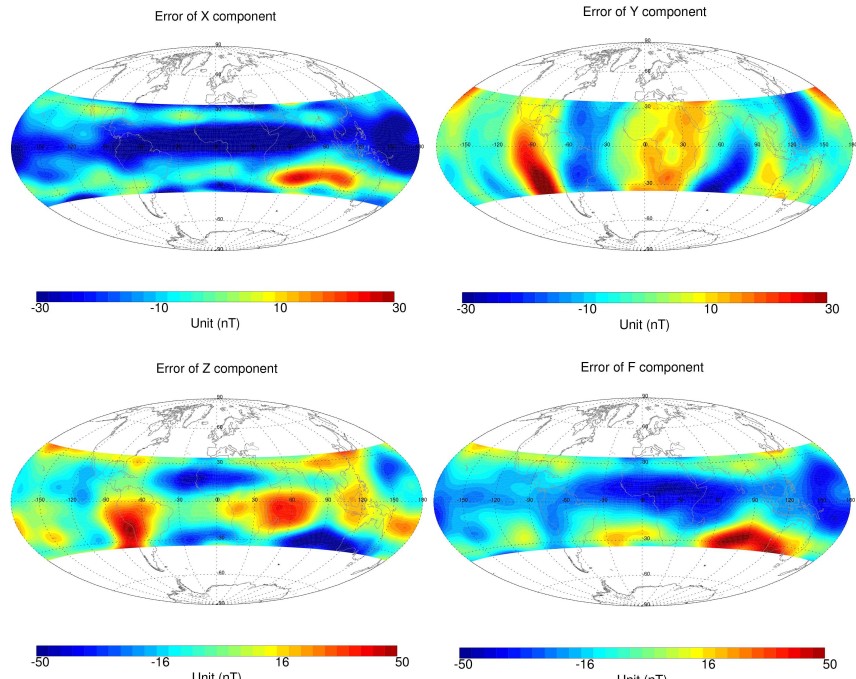

**Figure 14: The LOPS magnetic field residuals (after removing the core field giving by CHAOS-7 model) for three component and field intensity F. The map projection is Hammer-Aitoff.**




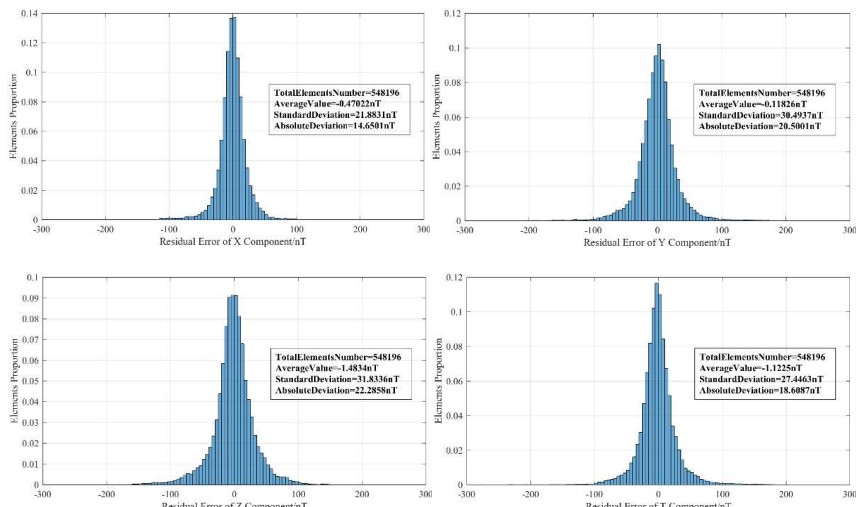

**Figure 15: A statistical distribution of LOPS Magnetic field vector and intensity residual for 1st April to 30th April, 2018 (after removing the core, crustal, and magnetospheric fields as given by CHAOS-7 model [Finlay et al., 2020]).**


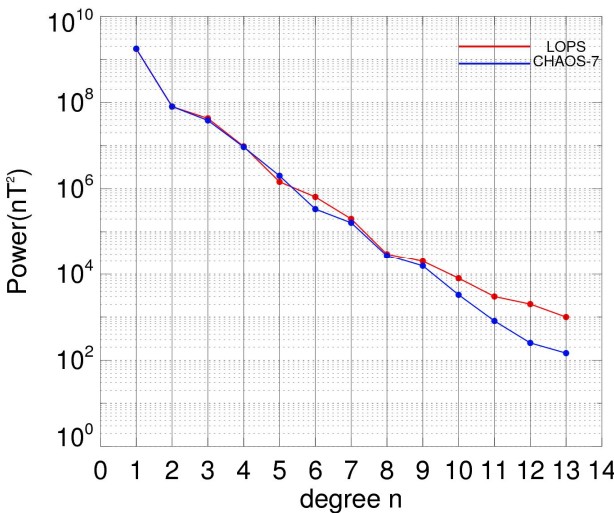

**Figure 16: Lowes–Mauersberger spherical harmonic power spectra up to degree n=13 of the vector magnetic field from both CHAOS-7 model (in blue) and LOPS data-based model (in red ) in April, 2018.**





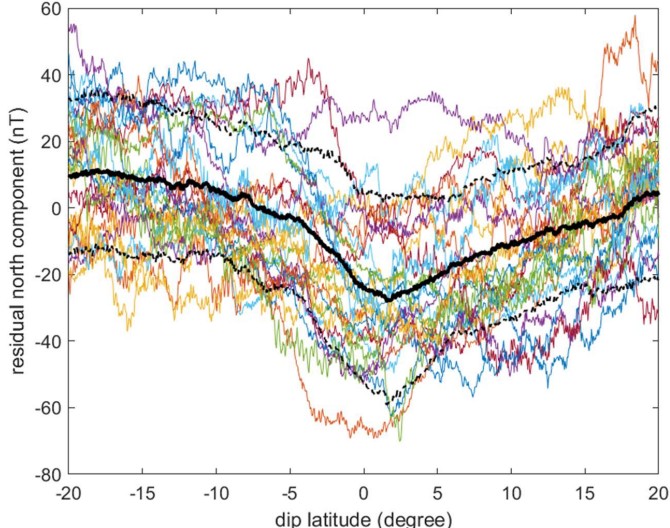

**Figure 17: The residual north components of different orbits after remove the core field, crustal field, and the magnetospheric field calculated from CHAOS-7 model. Different coloured lines indicate measurements from different orbits. The black thick line referred to the averaged values of all the orbits. Two black dashed lines indicate the standard deviation range.**