# Peer review of "The Fluxgate Magnetometer of the Low Orbit Pearl Satellites"

_Geoscientific Instrumentation, Methods and Data Systems, 2021_

## Author Response (AR1)

We thank you for both reviewers for their important comments. Below are the responses to the reviewer's comments point by point (in blue):

Referee 1

- Please, do not put acronyms in the abstract.
- Please, put the acronyms from the introduction, always the first time they appear and if they are to be used again.
- Line 24: correct the degree symbol and define LOPS.
- Line 45: correct the degree symbol.
* * *
Thank you for pointing out our typo errors, we make corresponding corrections in the revised manuscript.
* * *
- Line 65-67: you refer to the errors of the three Euler angles. Have these errors been evaluated?
* * *
Thank you for the comments. We add descriptions to the Euler angles in the revised manuscript.

The Euler angles, which describe the transformation from the magnetometer frame to the star imager frame, are usually estimated in two ways [e.g. Olsen, 2003; Olsen et al., 2006]:

$$\boldsymbol{B}_{VFM} = \mathbf{R} \cdot \mathbf{T} \cdot \boldsymbol{B}_{NEC} = -\mathbf{R} \cdot \mathbf{T} \cdot \nabla V \qquad (1)$$

Where $\boldsymbol{B}_{VFM}$ and $\boldsymbol{B}_{NEC}$ are the calibrated magnetic measurement in the magnetometer frame and geocentric frame, respectively. $\mathbf{R}$ and $\mathbf{T}$ are the transformation matrix determined by three Euler angles and attitude transformation matrix given by the Star Imager. In both of the ways, the estimation of the Euler angle requires both $\boldsymbol{B}_{VFM}$ and $\nabla V$. However, due to the absence of the absolute magnetic measurements, we could not determine the $\boldsymbol{B}_{VFM}$ directly. The information of the Euler angles are contained in the total compensation parameters $\boldsymbol{K}$ and $\boldsymbol{b}$ (see Equ. 16).
* * *
- Line 185: Please, can you add references on similar studies?
* * *
We add related references in the revised manuscript. The references are already cited in the Introduction section.

Previous studies have shown that the spacecraft stray field caused by static interference (DC field) or dynamic interferences could be detected and removed to be below the threshold of the scientific requirement by using a difference or gradient method based on dual sensors measurements. [e.g. Zhang et al., 2006, 2007; Auster et al., 2008; Ludlam et al., 2008; Pope et al., 2011].
* * *
● Line 228-232: This section could be better explained, step by step, adding graphs that better explain the process and indicating its references.
* * *
Thank you for the comments. In the revised manuscript, we explain the process in more detail.

It is possible that the sawtooth signal is associated with the loading current of the satellite. The current system of the satellite is quite complex so that we do not analyse the current itself but diminish its effect mathematically and obtain a reasonable background magnetic field. At first, we obtain the low-frequency components ($S_{spline}$) by making a smoothing spline fitting to the original data series. After that, we extract the high-frequency signal ($\Delta S$) by subtracting the low-frequency components from the original data series ($S$):

$$\Delta S = S - S_{spline} \qquad (2)$$

The sawtooth signals are then contained in $\Delta S$. The first order differences ($D_{\Delta S}$) of $\Delta S$ are calculated and divided into several segments with one segment about 100 data points. For each data segment of $D_{\Delta S}$, we determine the threshold above which we treat as outliers and set to be zero. The threshold is set empirically and varies from time to time but for most of the data, we set it to be a value below which there are 85% of the data points. We then obtain the calibrated high-frequency components by cumulative summation of the new first order difference ($D_{CAL}$):

$$\Delta S_{CAL} = CumSum(D_{CAL}) \qquad (3)$$

Finally, the calibrated data series are the summation of the calibrated-frequency components plus the low-frequency components.
* * *
● Line 285: References to Figure 16 and 17 appear before reference to Figure 15. Please, renumber the figures.
* * *
We renumber the Figures 16, 17 and Figure 15 in the revised manuscript.
* * *
Referee 2:

It is a nice paper, they show in a very careful way a fluxgate sensors calibration method. Their results in earth magnetic field measurement could be the beginning of further interesting results.

After mention the classic Howland Current Pump in my opinion they must include a citation as: T.I. AN-1515 report "*Comprehensive Study of the Howland Current Pump* "or something similar.
* * *
Thank you for the positive comments. In the revised manuscript, we cite the T.I. AN-1515 report "*A Comprehensive Study of the Howland Current Pump*" to illustrate the Howland current source circuit.

Howland Current Pump is excellent for put out a bidirectional current, and it can be used to force currents into sensors in production test [Robert A. Pease, T.I. AN-1515, 2008]. When the resistance is matched, the output resistance tends to be infinite. At this time, the voltage signal is converted into a linear current signal, which is independent of the load and the operating frequency, that is, the output current is a constant, and resulting a constant compensation magnetic field. We use "Improved Howland" current Pump, which can efficiently force as low as microamperes into voltages as large as 10 volts [Robert A. Pease, T.I. AN-1515, 2008].
* * *
Thank you very much
With Best Regards
Hao Luo
Institute of Geology and Geophysics, Chinese Academy of Sciences
P.O.Box 9825
Beijing 100029 P.R. China
Email: luohao06@gmail.com